# Twelve Months of Time-Restricted Feeding Improves Cognition and Alters Microbiome Composition Independent of Macronutrient Composition

**DOI:** 10.3390/nu14193977

**Published:** 2022-09-24

**Authors:** Abbi R. Hernandez, Cory Watson, Quinten P. Federico, Rachel Fletcher, Armen Brotgandel, Thomas W. Buford, Christy S. Carter, Sara N. Burke

**Affiliations:** 1Department of Medicine, Division of Gerontology, Geriatrics, and Palliative Care, University of Alabama at Birmingham, Birmingham, AL 35205, USA; 2Department of Neuroscience and McKnight, Brain Institute College of Medicine, University of Florida, Gainesville, FL 32610, USA; 3Birmingham/Atlanta Geriatric Research, Education, and Clinical Center, Birmingham VA Medical Center, Birmingham, AL 35205, USA

**Keywords:** cognitive aging, dual tasking, gut–brain-axis, intermittent fasting, metabolism

## Abstract

Declining health, gut dysbiosis, and cognitive impairments are hallmarks of advanced age. While caloric restriction is known to robustly extend the healthspan and alter gut microbiome composition, it is difficult maintain. Time-restricted feeding or changes in dietary macronutrient composition could be feasible alternatives for enhancing late life cognitive and physical health that are easier to comply with for extended periods of time. To investigate this possibility, 8-month-old rats were placed on time-restricted feeding with a ketogenic or micronutrient- and calorically matched control diet for 13 months. A third group of rats was permitted to eat standard chow ad libitum during this time. At 22 months, all rats were tested on a biconditional association task and fecal samples were collected for microbiome composition analysis. Regardless of dietary composition, time-restricted-fed rats had better cognitive performance than ad libitum-fed rats. This observation could not be accounted for by differences in motivation, procedural or sensorimotor impairments. Additionally, there were significant differences in gut microbiome diversity and composition between all diet conditions. Allobaculum abundance was associated with cognitive task performance, indicating a link between gut health and cognitive outcomes in aged subjects. Overall, time restricted feeding had the largest influence on cognitive performance in aged rats.

## 1. Introduction

Two prominent hallmarks of advancing age are declining peripheral health and impaired cognitive function, which can bi-directionally influence each other [1]. Caloric restriction, which has been shown to increase lifespan in several species, has been posited to increase healthspan and cognitive function [2] as well, although the data on the latter are equivocal [3]. The difficulty of maintaining long-term caloric restriction in humans, however, limits the translational potential of this lifestyle intervention for improving cognitive and physical function in older adults. Importantly, both time-restricted feeding (which is comparable to intermittent fasting) [4] and nutritional ketosis [5] mimic several aspects of caloric restriction and may confer health benefits to aged populations while imposing less severe dietary restrictions. Diet-based interventions may also alter the gut microbiome, which could directly influence brain function through the gut–brain axis, which is comprised of a multitude of pathways and interactions between the central nervous system and the gut. Specifically, ketogenic diets have been shown to influence gut microbiome abundance and diversity [6,7,8,9,10,11], which are also altered by advanced age [12]. 

A previous study reported that >3 months of nutritional ketosis, when initiated in aged rats (>21 months old), resulted in improved cognitive function on a biconditional association task compared to rats that were on a standard carbohydrate-based control diet. Importantly, both diets had equivalent caloric and micronutrient content and were given with a time-restricted feeding regimen once per day. Prior to initiating experimental diets, the aged animals in this previous study had metabolic impairments, including hyperinsulinemia and excess visceral white adipose tissue that was not evident in the young animals and reversed by nutritional ketosis, but not the standard diet after 4 weeks [13]. The possibility therefore exists that diet-induced cognitive benefits of ketosis are directly related to the efficacy of reversing metabolic deficits rather than due to the elevation of ketone bodies directly improving brain function. As the long-term carbohydrate restriction that is necessary to maintain nutritional ketosis poses huge barriers for extended compliance in community dwelling older adults [14], it is critical to determine the mechanisms by which this diet confers cognitive resilience. Different dietary paradigms that may increase compliance and therefore efficacy, such as time restricted feeding rather than restriction of either calories or macronutrient content. Specifically, if dietary interventions are initiated in mid-life, prior to the development of metabolic dysfunction, does ketosis still show the same benefits on cognitive function in old age compared to a standard diet. Importantly, mid-life is a critical time point for intervention, as this is likely when early pathology associated with sporadic Alzheimer’s disease and related dementias is detectable but has not yet produced cognitive impairment (e.g., [15,16]). 

The goal of this study was to therefore investigate whether long-term time-restricted feeding initiated in adulthood could improve cognitive outcomes in advanced age, and the extent to which this interacts with macronutrient composition. Two groups of rats were placed on a time-restricted feeding regimen beginning at 8 months of age. These rats were given ~51 kcal of food once daily. All animals consumed the full ration of calories within 3 h, resulting in ~21 h of fasting [13]. Among the rats given time-restricted feeding, one group was fed a ketogenic diet, while the other group was fed a micronutrient and calorically equivalent control diet [17]. A third group of rats was fed ad libitum until 21 months of age, at which time they were fed standard rodent chow once daily to encourage appetitively motivated participation in cognitive testing. A previous study has reported that rats of the Fischer 344 x Brown Norway hybrid strain develop hyperinsulinemia and metabolic impairments when allowed unrestricted access to standard laboratory rodent chow from adulthood into old age [13]. 

In old age, all rats were tested on a biconditional associated task (BAT), which quantifies an animal’s ability to cognitively multitask by simultaneously alternating between two different arms of a maze while completing a bi-conditional object discrimination. Specifically, the correct choice of the target object depends on the animal’s location on the maze with each object only being rewarded in one of the two arms (Figure 1). Performance on this type of object-place paired associative learning task has repeatedly been shown to decline with age in rats [18,19,20,21], and has greater sensitivity for detecting age-related impairments than the Morris watermaze test of spatial learning and memory [19]. Critically, the BAT is more comparable to complex cognitive tasks that older human adults must complete for instrumental activities of daily living and therefore is a better behavioral metric for assessing the translational potential of novel interventions. Potential confounds due to differences in motivation, or procedural and sensorimotor impairments, were assessed with a simple object discrimination problem, in which performance is typically not impaired in aged rats [18].

There is a plethora of mechanisms by which diet interventions could improve cognition in old age. The gut–brain-axis, or interaction between gut and brain health and function, has been recently identified as a powerful player in physiological functions in a variety of conditions. Moreover, gut dysbiosis, or a perturbation in the normal composition and/or density of the gut microbiome, is rampant with advanced age [12]. Cognitive deficits have been increasingly linked to changes in the gut microbiome [22,23] as well as alterations in metabolic function [24]. Therefore, we also investigated changes in the gut microbiome across rats fed these three different diets. While it is well established that ketogenic [6,7,8,9,10,11] and other [25] diets can significantly alter microbiome composition, to our knowledge, this is the first paper to investigate gut microbiome changes in response to TRF in combination with a ketogenic diet in an animal model, and the first to relate these changes to cognitive outcomes in old age. Furthermore, potential interactions between TRF and age-related changes in gut and metabolic health have also not yet been explored. 

## 2. Materials and Methods

### 2.1. Subjects and Dietary Interventions

33 aged (22 months) male Fisher 344 x Brown Norway F1 (FBN) Hybrid rats from the National Institute on Aging colony at Charles River were used in this study. All experimental procedures were performed in accordance with National Institutes of Health guidelines and were approved by Institutional Animal Care and Use Committees at the University of Florida. All rats were housed individually and maintained on a 12-h light/dark cycle with all behavioral testing occurring in the dark phase. Rats were divided into three groups: (1) fed ad libitum standard rodent chow until 21 months (*n* = 13), (2) fed 51 kCal of a standard diet once daily from months 8 to 21 (*n* = 10) and (3) fed 51 kCal of a ketogenic diet once daily from months 8 to 21 (*n* = 10). These group sizes were derived via power analysis utilizing a preliminary cohort of 3–4 rats per diet group through two-sample inference-estimation of sample size [26]. At 21 months of age, all rats were further restricted (approximately 25–30 kCal/day) to encourage participation in the appetitively motivated BAT behavior. Water was provided to all rats ad libitum throughout the study.

The same ketogenic diet (KD) and micronutrient matched control diet (CD) were used as published previously [17,20,27]. An additional group of rats were fed ad libitum with standard laboratory chow (Envigo, Teklad 2918). The KD was a high fat/low carbohydrate diet (Lab Supply; 5722, Fort Worth, TX, USA) mixed with MCT oil (Neobee 895, Stephan, Northfield, Illinois) with a macronutrient profile of 76% fat, 4% carbohydrates, and 20% protein. The micronutrient-matched CD (Lab Supply; 1810727, Fort Worth, TX, USA) had a macronutrient profile of 16% fat, 65% carbohydrates, and 19% protein. Nutritional ketosis was verified by testing peripheral levels of glucose and the ketone body β-hydroxybutyrate (BHB) 1 h after feeding. 

### 2.2. Behavioral Testing 

Rats were trained on the biconditional association task (BAT) as previously published [19,21,28]. Briefly, rats were first trained to alternate between left and right arms of a V-shaped maze (see Figure 1) with a macadamia nut reward at the end of each arm. Alternation training continued until rats reached a criterion performance of ≥80% correct with completion of all 32 trials within 20 min. Failing to alternate was logged as an incorrect trial. Following alternation training, rats began testing on the BAT, in which a single object pair was placed over two different food wells in the choice platform at the end of both arms (Figure 1B, orb and blue poodle). One object covered a hidden food reward that the rat could retrieve for moving the correct object. Importantly, in the left arm the orb was the rewarded object while in the right arm the poodle was the rewarded object. Rats were allowed to eat the food reward (macadamia nut) if they correctly displaced the object contingent on the current location within the maze. Rats were given 32 trials per day in alternating arms with objects placed pseudorandomly on the left and right sides within a given arm. Rats were trained until a criterion performance of ≥80% correct for each object on 2 consecutive days. Following criterion performance on the BAT, all rats were tested on a simple object discrimination within a single arm of the maze. For this control task, object choice was not contingent upon location, and the same object was always rewarded. For both tasks involving objects, selecting the unrewarded object was logged as an incorrect trial. During this phase of testing, rats did not make any alternation errors. 

### 2.3. Statistical Analysis

All data are expressed as group means ± standard error of the mean (SEM) unless otherwise reported. Glucose ketone index (GKI), body weights during behavior, and behavioral performance on all tasks were analyzed using a one-way ANOVA across diet groups. For all behavioral tasks, outliers were determined using the ROUT method [29] with a false discovery rate of 0.1 prior to ANOVA, and normality was determined with the omnibus K2 test. One outlier was detected in the ketogenic-fed rats during alternation training, one outlier was detected in the ketogenic-fed rats during WM/BAT training and one ab lib and two control-fed rats were outliers during the object discrimination task. Body weight throughout the duration of the study was analyzed using repeated measures-ANOVA (RM-ANOVA) across diet groups. When applicable, follow up comparisons were performed between individual groups using t-tests adjusted with Bonferroni’s multiple comparisons test. Finally, to examine for potential relationships across variables, a principal component analysis (PCA) was performed with a varimax rotation. Factors with eigenvalues above 1.0 were considered meaningful and loading coefficients below 0.50 were excluded as done previously [18]. 

### 2.4. Fecal Microbiome Taxonomy 

At the time of sacrifice, fecal samples were collected from 9 CD, 9 KD and 12 ad *lib-fed* rats, directly from the distal colon. Samples were immediately placed in Para-Pak (Meridian Bioscience Inc., Cincinnati, OH), frozen on dry ice and stored at −80 °C until use.

Samples were processed by the Microbiome Institutional Research Core at the University of Alabama at Birmingham using previously published methods [30,31,32]. Briefly, analysis of fecal microbiome was performed via 16S rRNA gene sequencing. Amplicon sequence variants (ASVs) were resolved and taxonomy was assigned using the SILVA small subunit ribosomal RNA database version 132 [33]. Alpha diversity was calculated utilizing the microbiome package in R [34], and beta diversity was calculated utilizing the Phyloseq package in R [35] via permutational multivariate analysis of variance (PERMANOVA). Analysis of Compositions of Microbiomes (ANCOM) with Bias Correction was used to test for differential using modified versions of previously published ANCOM scripts with a detection limit of 0.7 [32,36,37]. 

## 3. Results

### 3.1. Peripheral Effects of Feeding Paradigms

Postprandial glucose (Figure 2A) and BHB (Figure 2B) measurements were utilized to generate a glucose ketone index (GKI) for each rat as reported previously [27]. Lower GKI values indicate greater levels of ketosis. GKI values during behavioral testing indicate only rats fed the ketogenic diet were in nutritional ketosis (F_[2,29]_ = 54.07; *p* < 0.001; Figure 2C). There was no significant main effect of feeding method (F_[1,30]_ = 0.76; *p* = 0.39), as rats on time-restricted feeding with the standard diet had a significantly elevated GKI level relative to ad libitum-fed rats (t_[29]_ = 4.18; *p* < 0.001), but rats on time-restricted feeding with the ketogenic diet had a significantly lower GKI level relative to ad libitum-fed (t_[29]_ = 6.87; *p* < 0.001).

Over the course of time-restricted feeding, both ketogenic and control-fed rats gained a significant amount of weight (F_[1,18]_ = 378.50; *p* < 0.001), indicating that they were not calorically restricted. Although caloric intake was identical, the control-fed rats gained significantly more weight that the ketogenic diet-fed rats (F_[1,18]_ = 7.08; *p* = 0.02). Moreover, the interaction between time and diet group was significant (F_[1,18]_ = 67.09; *p* < 0.001), indicating that the control-fed rats also gained weight more rapidly. Although this observation suggests that rats were not calorically restricted, weights did not reach the level of ad libitum-fed animals of the same age (Figure 2D). The control and ketogenic intermittent fasting groups reached maximal weights that were on average 472 and 421 g, respectively, compared to an average maximal weight of 547 g in the ad libitum-fed rats. This is likely because rats with unrestricted access to food overconsume, which could be related to the excessive visceral fat and metabolic impairments that have been reported for aged male rats of this strain [13,17].

Prior to beginning shaping, all rats were placed on food restriction to motivate appetitive behavior for cognitive testing. Animals were given ~25-30 Kcal/day (black arrow in Figure 2D), which initiated weight loss in all groups. Body weight on the first day of alternation and BAT testing (dashed arrow in Figure 2D) was significantly lower in ketogenic diet-fed rats than in control-fed (t_[30]_ = 5.00; *p* = 0.004; t_[30]_ = 5.43; *p* = 0.002, respectively) and ad libitum-fed rats (t_[30]_ = 7.15; *p* < 0.001; t_[30]_ = 4.01; *p* = 0.02, respectively), though the control-fed and ad libitum-fed rats did not significantly differ (t_[30]_ = 1.83; *p* = 0.41; t_[30]_ = 1.76; *p* = 0.44; respectively Figure 2D,E). The comparable body weights during behavioral testing between the time-restricted control-fed rats and rats that ate ad libitum between 8 and 21 months suggest that potential differences in behavior cannot be explained by differences in overall body condition.

### 3.2. Time Restricted Feeding, Regardless of Macronutrient Composition, Ameliorated Age-Related Cognitive Impairment on the Biconditional Association Task

The total number of incorrect trials across all days of training through the final day of criterion performance for all tasks were tabulated for each rat. For alternation training, neither diet group (ketogenic versus standard; F_[2,29]_ = 1.51; *p* = 0.24) nor feeding method (ad libitum versus TRF; F_[1,29]_ = 0.12; *p* = 0.73) significantly affected the number of incorrect trials required to reach criterion performance on alternations throughout the maze (Figure 3A). Aged rats typically perform comparable to young on this behavior [19]. The similar performance accuracy on alternations across diet groups indicates that despite differences in weight, rats in all diet and feeding groups were similarly motivated to retrieve the food reward.

The number of incorrect trials required to reach criterion performance on BAT testing was significantly different across the diet groups (F_[2,28]_ = 4.48; *p* = 0.02; Figure 3B). These results were due to the time-restricted feeding from young adulthood rather than dietary macronutrient composition, as the method of feeding also had a significant effect on performance (F_[1,30]_ = 194.07; *p* < 0.001). Ad libitum fed rats required significantly more trials to reach criterion performance than ketogenic diet-fed rats (t_[1,28]_ = 2.63; *p* = 0.04) and a strong trends towards significantly more trials than control-fed rats (t_[1,28]_ = 2.38; *p* = 0.07). Conversely, among the rats that were given time-restricted feeding from young adulthood into old age, there was no difference in performance between the ketogenic and control diet groups (t_[28]_ = 0.22; *p* > 0.99).

Data from the cohort presented here were also compared to young animals from other previously run cohorts that underwent identical BAT behavioral testing to assess how life-long TRF influenced cognitive outcomes in advanced age. While old rats made significantly more incorrect trials than young prior to reaching criterion performance on BAT testing (F_[1,64]_ = 19.22, *p* < 0.001), TRF-fed rats overall performed significantly better than ad lib fed rats regardless of age group (F_[1,64]_ = 7.64, *p* < 0.01). Moreover, when the ketogenic and control groups of TRF fed rats were compared to young rats, there were was no significant difference in performance across the three groups (F_[2,38]_ = 1.35, *p* = 0.27). In contrast, the ad libitum-fed aged rats made significantly more errors than the young rats (T_[39]_ = 4.40, *p* = 0.001). Together these data indicate that a mid-life intervention with TRF may prevent age-related cognitive deficits commonly observed on this task.

Following BAT testing, a simple object discrimination control was utilized to ensure all rats were able to discriminate between two dissimilar objects and to assess potential differences in motivation or procedural impairments (Figure 3C). There was not a significant difference in the number of incorrect trials required to reach a criterion performance on this control task across the 3 groups of rats (F_[2,25]_ = 2.60; *p* = 0.10), nor was there an effect of feeding method (ad libitum versus time-restricted; F_[1,29]_ = 0.82; *p* = 0.37). These data demonstrate all rats were able to discriminate between objects and are not visually impaired, indicating BAT task performance was not hindered by physical or sensory deficits, but rather was likely to manifest from cognitive differences across diet groups that altered the rate at which animals could learn the object-in-place rule.

### 3.3. Time Restricted Feeding and the Ketogenic Diet Influence Gut Microbiome Composition and Beta Diversity

At the phylum taxonomic level (Figure 4A,B), two phyla were found to significantly differ across diet groups: actinobacteria (F_[2,27]_ = 11.49; *p* < 0.001) and deferribacteres (F_[2,27]_ = 7.39; *p* = 0.002). While both TRF groups had significantly reduced abundance of actinobacteria relative to the ad libitum fed group (*p* < 0.01 for both comparisons), only the ketogenic diet fed TRF group had significantly higher levels of deferribacteres than the ad libitum fed group (t_[27]_ = 3.30; *p* = 0.006; Figure 4C) but not the control TRF fed group (t_[27]_ = 0.38; *p* > 0.99). No other phyla differed by diet group or feeding paradigm (*p* > 0.19 for all comparisons), nor did the ratio of firmicutes to bacteroidetes differ (*p* > 0.30 for both comparisons; Figure 4E). In addition to broad changes at the phylum level, additional analyses (see ANCOM below) were conducted at the genus level (Appendix A) to assess more detailed differences in gut microbiome composition.

Two common measures of alpha diversity, the Inverse Simpson (IS) index [40], which measures the dominances of a multispecies community [40] and Shannon’s (S) index, which takes taxa richness into account [41], were utilized. There were no differences across groups in alpha diversity by either measure, whether separated by dietary macronutrient composition (S: F_[2,27]_ = 0.37, *p* = 0.72; IS: F_[2,27]_ = 1.32, *p* = 0.28; A,B) or by feeding paradigm (S: t_[28]_ = 0.67, *p* = 0.51; IS: t_[28]_ = 1.58, *p* = 0.13; Figure 5C,D).

Beta diversity was calculated using Bray Curtis (BC) Dissimilarity and differences across groups were assessed with a permutational multivariate analysis of variance (PERMANOVA). PERMANOVA revealed a significant effect of both diet (F_[2,29]_ = 4.87; *p* = 0.001) and feeding paradigm (F_[1,29]_ = 5.81; *p* = 0.001) on beta diversity. As beta diversity has been shown to correlate with better cognition in middle-aged adults [42], this is one potential mechanism by which intermittent fasting could improve BAT performance accuracy.

Analysis of composition of microbiomes (ANCOM) was utilized to examine taxa that had statistically different abundance between diet and feeding paradigm groups (Figure 6). Interestingly, different phyla and genera were identified by the two analyses. ANCOM across diet groups at the phylum level revealed 2 significantly different taxa, deferribacteres and Euryarchaeota. At the genus level, 4 taxa were differentially abundant across diet groups, 3 of which were in the firmicutes phyla (*Ileibacterium*, *[Ruminococcus] gauvreauii group*, and *Allobaculum*) and 1 from the tenericutes phyla (an uncultured bacterium).

ANCOM across the different feeding paradigms identified only one significant difference at the phylum level, Actinobacteria. At the genus level, the same uncultured tenericutes group and *Allobaculum* from the firmicutes phyla were significantly different across feeding paradigm, along with two other firmicutes microbiota *(Intestinimonas* and *[Eubacterium] ventriosum group).*

### 3.4. Alterations in Gut Microbiome Composition Correlate with Behavioral Performance

A principal component analysis (PCA) was used to determine if there was a relationship between the significantly altered gut microbiota identified by ANCOM (see above) with behavioral performance and biomarkers of ketosis (Figure 7). Three components had eigenvalues > 1, which combined accounted for 61.60% of the variance.

The first component, which accounts for 27.72% of the total variance with an Eigenvalue of 2.77, loaded negatively with circulating ketones (−0.87) and negatively with circulating glucose (0.82). Thus, component 1 largely represents dietary ketosis during behavioral task performance. Moreover, this component loaded negatively with the genera *Intestinimonas* (−0.66) and positively with *Ileibacterium* (0.61) and *Ruminococcus gauvreauii group* abundance (0.52), demonstrating a relationship between abundance of these genera with a ketogenic diet. The second component, which accounts for 20.31% of the total variance with an Eigenvalue of 2.03, positively correlates with the number of errors made on BAT prior to reaching criterion (0.73) and *Allobaculum* abundance (0.79). Thus, this component suggests that worse cognition was associated with greater abundance of a specific genera that is modified by both diet composition and feeding paradigm. The third component, which accounts for 13.58% of the total variance with an Eigenvalue of 1.36, positively loaded with *Eubacterium ventriosum group* (0.71) and the uncultured tenericutes bacterium (0.77), and negatively with *Ruminococcus gauvreauii group* abundance (−0.50).

## 4. Discussion

The data presented here show that time-restricted feeding initiated in mature adults, prior to the onset of age-related metabolic impairments, can influence cognitive outcomes in advanced age. Specifically, rats fed once daily between the ages of 8 and 21 months with either a ketogenic or a standard control diet performed better on a cognitive biconditional association task (BAT) compared to rats that were fed ad libitum during that same time frame. This observation could not be accounted for by differences in body weight or sensorimotor impairments. It is important to note than although rats were fed once daily, and thus prevented from obesogenic overconsumption typically observed with ad libitum feeding [43], these rats were not calorically restricted and continued to show modest increases in weight throughout their lives.

Furthermore, we observed that nutritional ketosis, which has been shown to improve metabolic health in aged rats [13,17], did not confer an additive benefit to time-restricted feeding with a standard control diet in regard to multitasking performance on a biconditional association task. Shorter term ketogenic diets initiated in old age (21 months), which is an age at which rats on lifelong ad libitum feeding have age-related declines in metabolic function, have demonstrated improved cognitive outcomes on a similar task as well as reduced anxiety-like behavior [20]. An important distinction between these two studies is the timing of the initiation of ketogenic diet therapy. When initiated at 8 months of age, normal rats show little metabolic dysfunction from lifelong ad libitum feeding with a standard diet. In contrast, rats that are allowed to consume food ad libitum into old age gain excessive weight, acquire aberrant amounts of white adipose tissue, as well as show disrupted insulin signaling and a reduced ability to utilize glucose in the brain [13,17,43,44]. The current data suggest that time-restricted feeding throughout adulthood, regardless of the macronutrient composition, may be able to prevent these metabolic deficits in old age and lead to resilience against age-related cognitive decline. In contrast, when diet interventions are initiated in old age, declines in metabolic function need to be reversed. A previous study reported that time-restricted feeding with a ketogenic diet may be more effective at normalizing metabolic function in aged rats than time-restricted feeding with a standard diet. Thus, it may be critical to consider an individual’s current metabolic status when designing an optimal diet-based intervention for optimizing cognitive performance. This type of precision medicine-based approach has recently been suggested as an new avenue for treating cognitive aging [45].

The limited number of studies that have investigated the potential of TRF to alter gut microbiome composition have reported conflicting results. One group found significant changes in the microbiome following TRF in young healthy male human subjects [46]. Moreover, such changes correlated with changes in metabolic markers and circadian rhythm-associated genes [47]. However, in another study of obese humans, TRF did not significantly alter microbiome abundance or diversity, despite significant weight loss [39]. Here, we found significant changes from TRF in rats, as well as from altered macronutrient composition via a KD. Specifically, 2 firmicutes genera (*Ileibacterium* and *ruminococcus gauvreauii group*) were reduced in abundance by the ketogenic diet alone. When the microbiota from TRF (both ketogenic and control) were compared to ad lib fed rats 2 firmicutes genera (*Intestinimonas* and *Eubacterium ventriosum group*) were modified. One firmicutes genus (*Allobaculum*) was reduced in the TRF rats while and 1 tenericutes genus (an uncultured bacterium) was significantly elevated in abundance in the TRF compared to ad lib fed rats.

Not only do our data indicate both dietary macronutrient composition and feeding paradigm influence microbiome composition, but also that these changes correlate with behavioral performance. Specifically, worse performance accuracy on the biconditional association task was associated with higher *Allobaculum* abundance. *Allobaculum*, which is involved in butyric acid production [48,49], increases following the consumption of a high fat diet in rodents that produces cognitive impairments [38,50,51]. While this suggest that increased levels of *Allobaculum* may be related to suboptimal health, other studies have reported that voluntary exercise can lead to enhanced abundance of *Allobaculum* [52,53]. Thus, future studies should attempt to manipulate *Allobaculum* abundance directly to examine the impact on cognitive function and metabolic health.

The current data also have important implications in mid-life food consumption patterns and later life cognition. While is it well documented that high fat/high sugar obesogenic diets are associated with worse cognition [54], these data show that over consuming even a standard diet that does not contain high fat or excessive sugar can also lead to worse to cognitive performance later in life relative to alternative feeding styles. Thus, it is conceivable that adults who overconsume throughout mid-life are at a higher risk for cognitive decline in advanced age. A 2020 report by the U.S. Department of Health and Human Services found 17.5% of individuals 45–64 had diabetes [55], demonstrating the dire necessity of interventions during this critical period to avoid further cognitive decline in geriatric populations.

Our data provide additional support for the strong link between gut and brain function, known as the gut–brain-axis. However, what remains unclear is how alterations in diet, such as those performed in this study, are able to influence cognitive function. One potential explanation is that alterations in gut microbiome composition are capable of improving other aspects of systemic health, including restoring insulin-related signaling and preventing or decreasing inflammation. Moreover, changing the gut microbiome composition can significantly influence metabolite production and bioavailability, which can influence neurobiological processes. For example, one group found that diabetes-induced cognitive impairments were ameliorated following both fasting-induced changes in gut microbiome composition and direct application of affected metabolites [56]. Changes such as these then allow for the restoration of, or prevention of decline in, neurobiological function through related avenues, including nutrient availability and decreased inflammatory processes.

A second way by which gut health and gut microbiome composition influence cognitive outcomes is through a more direct impact on neurobiological function. This can take place through a variety of means, including altered levels of neurotransmitters produced by gut microbes and direct interaction of these microprobes and their metabolites with the enteric nervous system. To test this theory, experiments directly influencing these particular aspects of gut and nervous system function can help elucidate specific pathways to utilize as therapeutic targets. This can include things like a vagotomy, or the removal of part of the vagus nerve, to significantly sever the connection between gut function and the nervous system prior to dietary intervention. Alternatively, to decrease peripheral health-improving effects of dieting, such as weight loss or improved insulin resistance, fecal matter transplants from specific populations can supplemented to study more direct effects of gut microbiome population contributions to cognitive outcomes.

Moreover, these potential avenues through which gut composition and function influence neurobiological function may be overlapping and synergistic. It is likely that changes in one organ system has profound impacts on other organ systems, such as central nervous system function. Additionally, while the work presented here utilizes a well-published behavioral task that is sensitive to early cognitive decline, further cognitive characterization would complement our findings and enhance our ability to link peripheral function with cognitive performance. 

## 5. Conclusions 

Our data strongly suggest that interactions with peripheral and systemic functioning are important aspects to consider with interventional therapies targeting age-related cognitive decline through peripheral means, like through diet or orally ingested compounds. Thus, these data are in support of utilizing the gut microbiome as modifiable therapeutic target for alleviating cognitive dysfunction, which may occur through altered dietary macronutrient consumption, altered feeding patterns or other supplements such as pre- or probiotics.

## Figures and Tables

**Figure 1 nutrients-14-03977-f001:**
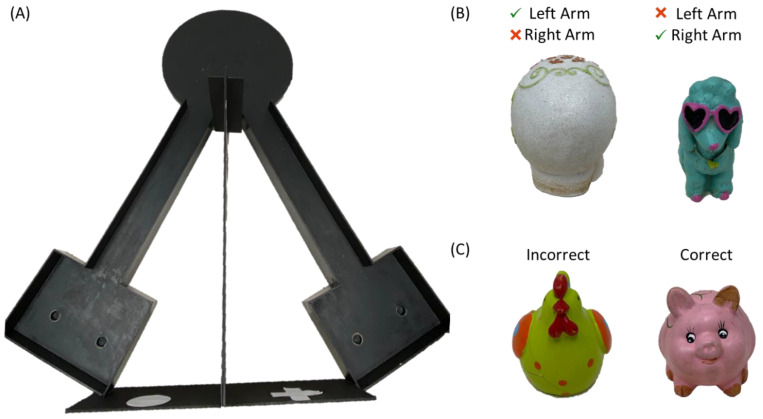
Behavioral testing apparatus and objects utilized for BAT and object discrimination tasks. (**A**) Birds-eye view of the testing apparatus used for alternations, BAT, and object discrimination tasks. Note during the object discrimination, only one arm of the maze was used, and rats were not required to alternate. Objects utilized during (**B**) BAT. In the left arm of the maze the white skull was rewarded when selected while the blue poodle was the correct in the right arm. Thus, the correct response updated based on spatial location. Objects used for (**C**) simple object discrimination testing. The pig was the correct choice.

**Figure 2 nutrients-14-03977-f002:**
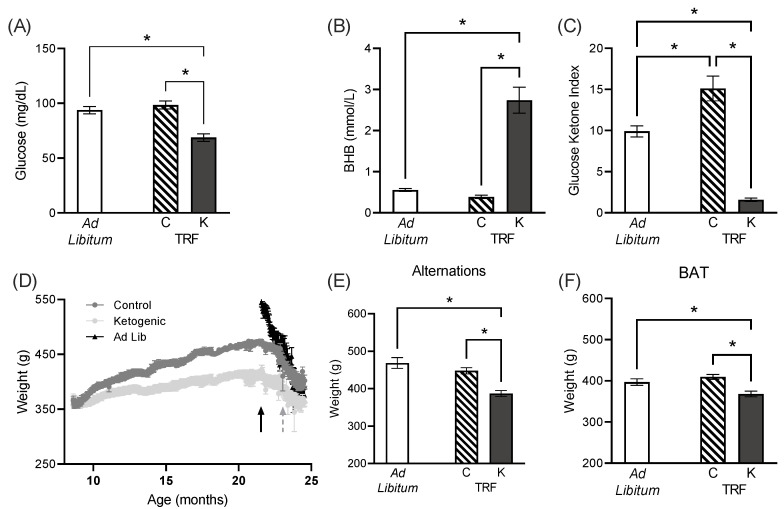
Postprandial glucose, BHB and GKI values and body weight. Only rats fed a ketogenic diet exhibited nutritional ketosis as evidenced by (**A**) reduced glucose, (**B**) elevated levels of the ketone body β-hydroxybutyrate and (**C**) a lower glucose ketone index (GKI). (**D**) Body weights throughout the lifespan continued to increase while fed a ketogenic or control diet via time-restricted feeding, until the onset of further dietary restriction (solid black arrow) and during BAT testing (beginning at dashed gray arrow). Body weight on the first day of (**E**) alternation training, as well as (**F**) BAT, was significantly lower in ketogenic diet-fed rats relative to both other groups. C, control-fed; K, ketogenic diet-fed; TRF, time restricted-fed; BAT, biconditional association task. All data are group means ± 1 SEM; * indicates *p* < 0.05.

**Figure 3 nutrients-14-03977-f003:**
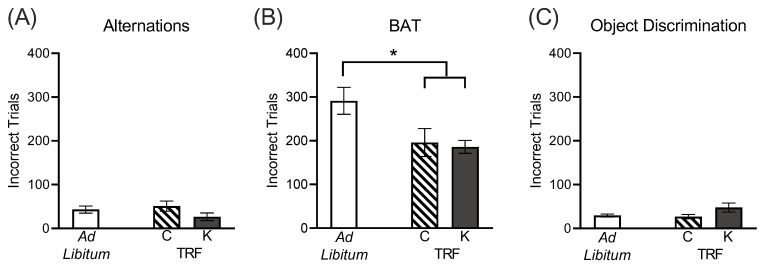
Behavioral performance of aged rats across diet groups and feeding methods. (**A**) There were no differences across groups in ability to alternate between the left and right arms of the maze. (**B)** Lifelong time-restricted feeding, regardless of macronutrient composition, improved ability to acquire the object-in-place rule required for criterion performance on the BAT task to a degree that was comparable to performance in young (4–8 mo) rats of the same strain. (**C)** All rats were able to perform an object discrimination task similarly, indicating no sensorimotor deficits or motivation differences across groups that would confound BAT task performance. All data are group means ± 1 SEM; * indicates *p* < 0.05.

**Figure 4 nutrients-14-03977-f004:**
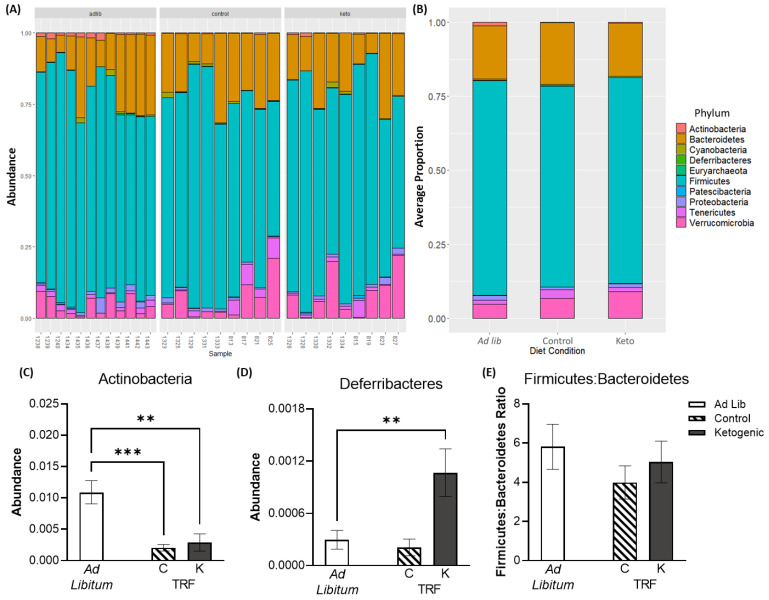
Relative abundance at the phylum taxonomic level by (**A**) subject and (**B**) diet condition. Relative abundance of (**C**,**D**) significantly altered phyla and (**E**) the ratio of firmicutes to bacteroidetes. All data in panels (**C**–**E**) are group means ± 1 SEM; ** indicates *p* ≤ 0.01; *** indicates *p* ≤ 0.001.

**Figure 5 nutrients-14-03977-f005:**
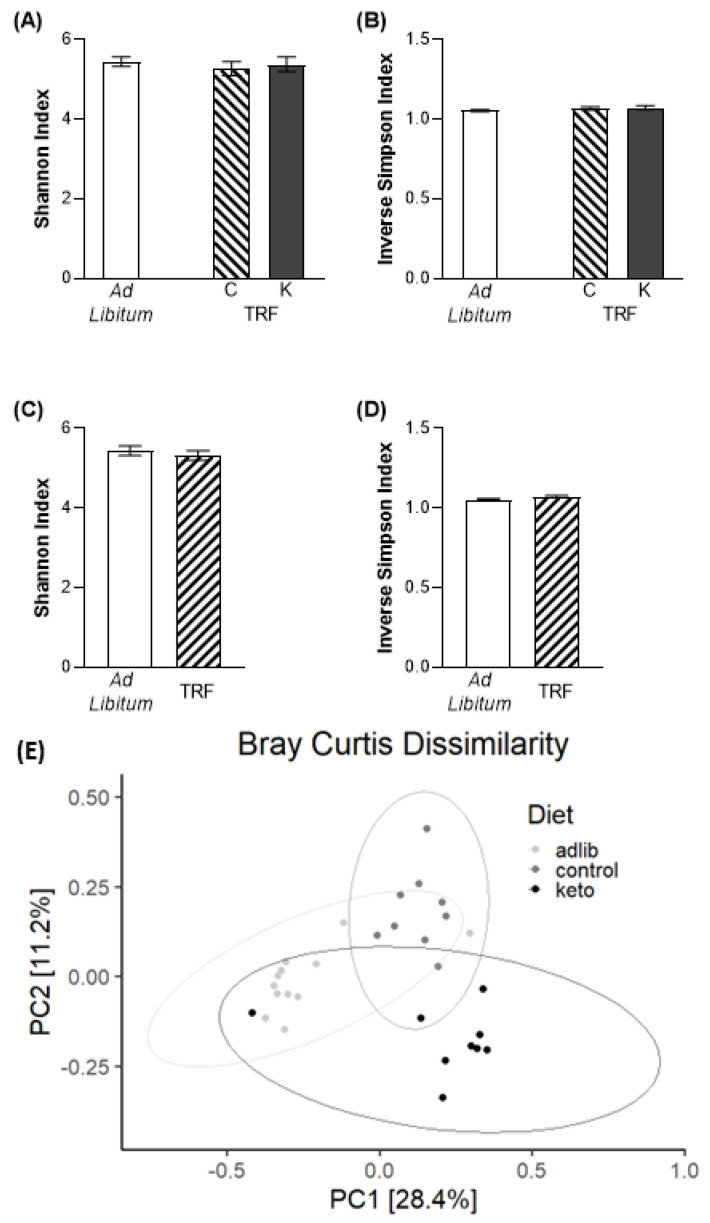
Beta, but not alpha, diversity differed across dietary paradigms. Neither the Shannon or Inverse Simpson indices of alpha diversity differed across (**A**,**B**) macronutrient or (**C**,**D**) feeding paradigm groups. (**E**) Conversely, beta diversity was different across both diet groups and feeding paradigms. Data in (**A**–**D**) are group means ± 1 SEM; PC = principal component.

**Figure 6 nutrients-14-03977-f006:**
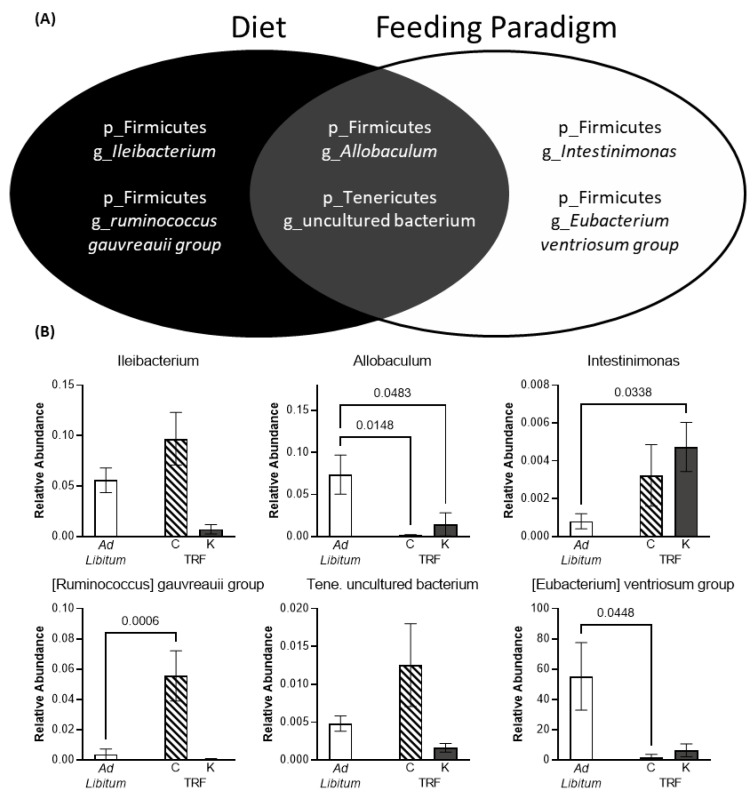
Significantly altered genera identified utilizing the analysis of composition of microbiomes (ANCOM) methodology. (**A**) 6 genera were identified to be significantly different across the different diet and/or feeding paradigm groups. (**B**) Relative abundance of the identified genera. All data are group means ± 1 SEM.

**Figure 7 nutrients-14-03977-f007:**
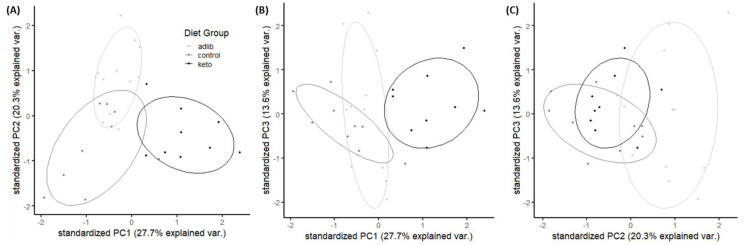
Principal component analysis on measures of nutritional ketosis, cognitive performance and microbiome composition. (**A**) Components 1 and 2, (**B**) components 1 and 3 (**C**) and components 2 and 3 are shown as a function of diet group (light gray = ad libitum fed rats, medium gray = control TRF fed rats and black = ketogenic diet TRF fed rats).

## Data Availability

Data available upon request.

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
