# Peer review of "Twelve Months of Time-Restricted Feeding Improves Cognition and Alters Microbiome Composition Independent of Macronutrient Composition"

_nutrients, 2022, doi:10.3390/nu14193977_

Round 1

Reviewer 1 Report

Time-restricted feeding or changes in dietary macronutrient composition could be feasible alternatives for enhancing late life cognitive and physical health. The present manuscript addresses the effect of time-restricted feeding on the improvement in cognitive performance and the composition of the microbiome.

Although the topic seems to be interesting per se the manuscript exhibits several shortcomings:

1.      A major problem with this study is that the authors performed only one behavioral experiment to illustrate the improvement in cognitive function. In my opinion, only BAT testing seem totally inadequate to evaluate the cognitive function.

2.      Another problem with this paper is that in terms of behavioral testing, the sample size is rather low.

3.      The authors do not have a good explanation for how changes in gut microbiome composition correlate with behavioral performance.

4.      The readers cannot extract useful information from Figure 4 and 5 as the figure quality is too poor.

Author Response

Reviewer 1: Time-restricted feeding or changes in dietary macronutrient composition could be feasible alternatives for enhancing late life cognitive and physical health. The present manuscript addresses the effect of time-restricted feeding on the improvement in cognitive performance and the composition of the microbiome. Although the topic seems to be interesting per se the manuscript exhibits several shortcomings:

  1. A major problem with this study is that the authors performed only one behavioral experiment to illustrate the improvement in cognitive function. In my opinion, only BAT testing seem totally inadequate to evaluate the cognitive function.
    1. The BAT was chosen to evaluate cognitive performance as it has been demonstrated to be a behavioral paradigm with high discriminative validity for detecting early stages of age-related cognitive decline (Hernandez et al., 2015). Moreover, performance on this task requires intact function of several key brain regions known to decline in function with age, including the hippocampus, medial prefrontal cortex and perirhinal cortex (Jo and Lee, 2010; Hernandez et al., 2017). Lastly, these animals were also tested on the Morris water maze (data not shown), but in accordance with other previous findings from rats of this strain and age, relative to young animals, there were no deficits in the ability to find the hidden platform on this task in any of the diet groups, and therefore the data were not informative. We believe this further demonstrates the utility of the BAT, as it was able to detect cognitive impairments prior to detectable water maze impairments. However, we have addressed this as a study limitation in the discussion section of the manuscript, as an additional cognitive battery would certainly enhance the ability to mechanistically link cognition with the peripheral health changes reported herein.
  2. Another problem with this paper is that in terms of behavioral testing, the sample size is rather low.
    1. The sample size was determined in two ways. Firstly, several previous studies have found significant group differences utilizing 5-12 rats/group on the same or highly similar task (Hernandez et al., 2017, 2018, 2020b, 2020a). Secondly, sample size was determined by utilizing a preliminary cohort of 3-4 rats/diet group with the Two-Sample Inference - Estimation of sample size and power described in Bernard Rosner's Fundamentals of Biostatistics, which revealed a sample size of 9 would be sufficient for each group to provide adequate statistical power to reject the null hypothesis of no effect of diet with 80% probability. Finally, our by Institutional Animal Care and Use Committees (IACUC) requires that we reduce the number of experimental subjects as much as possible in accordance with the 3 Rs (i.e., Replacement, Reduction and Refinement) from the NIH (Hubrecht and Carter, 2019). Thus, it would be problematic to obtain approval to run more animals after statistical significance had been achieved. Moreover, maintaining subjects on these diets, which were individually fed to each rat every single day for over a year, isn’t a feasible use of resources in a larger cohort. For all of these reasons combined, 10 rats were utilized per TRF group and 13 for the ad lib group to account for potential animal loss, as aged animals tend to have a high attrition rate. We have amended the methods section to include that we conducted the power analysis to justify animal numbers.
  3. The authors do not have a good explanation for how changes in gut microbiome composition correlate with behavioral performance.
    1. Within the discussion section, we propose that either the gut and brain are modulated by the dietary interventions separately, or that the diet modulates the gut microbiome composition which then alters neurobiological function through a host of possible mechanisms. These mechanisms could be directly modulating nervous system function through things like the enteric nervous system or altered neurotransmitter levels in the gut or through altered metabolic processes, such as changes in fatty acid bioavailability through the organism. This is discussed in lines 427-451 of the discussion.
  4. The readers cannot extract useful information from Figure 4 and 5 as the figure quality is too poor.
    1. Figures 5 & 7 have been amended (increased font size and resolution) for clarity. Moreover, we have removed panels G & F from figure 4, as we felt they did not significantly add value to the main manuscript, and have included them (in a larger, more readable size) as a supplemental figure.

Reviewer 2 Report

Well written article

Author Response

Thank you. 

Reviewer 3 Report

Comments:

1. Could the authors please provide animal ethical review board approval for their study and what steps were taken to minimize the use of animals please?

2. How were the numbers of animals determined i.e. how was the study powered?

3. Could the authors please justify the use of parametric statistical analyses please?

Author Response

Reviewer 3:

  1. Could the authors please provide animal ethical review board approval for their study and what steps were taken to minimize the use of animals please?
    1. We have amended the methods section to include the following statement: “All experimental procedures were performed in accordance with National Institutes of Health guidelines and were approved by Institutional Animal Care and Use Committees at the University of Florida.”
  2. How were the numbers of animals determined i.e. how was the study powered?
    1. A preliminary cohort of 3-4 rats/diet group was utilized to determine the required sample size for this study. Two-Sample Inference - Estimation of sample size and power, as described in Bernard Rosner's Fundamentals of Biostatistics, revealed a sample size of 9 would be sufficient for each group. Due to the high attrition rate of aged animals, 10 rats were utilized per TRF group and 13 for the ad lib group to account for potential animal loss. We have amended the methods section to include that we conducted this power analysis.
  3. Could the authors please justify the use of parametric statistical analyses please?
    1. Firstly, the omnibus K2 test (D'Agostino-Pearson normality test) was utilized to assess normality, which revealed all behavioral data was normally distributed. We have amended the methods section to include this statistical test. Secondly, the groups had unequal variances, which makes parametric analysis more appropriate.

Round 2

Reviewer 1 Report

Thanks for your professional responses.

Reviewer 3 Report

The authors have satisfactorily addressed my comments.